# The Influence of Viewing Photos of Different Types of Rural Landscapes on Stress in Beijing

**Xiaobo Wang [1,*], Hanlun Zhu [2], Zhendong Shang [1] and Yencheng Chiang [3]**

1   College of Architecture and Art, North China University of Technology, Beijing 100144, China; szd_bl@163.com
2   Orient Landscape, Beijing 100015, China; zhuhanlun@orientscape.com
3   Department of Landscape Architecture, National Chiayi University, Chiayi City 60004, Taiwan; ycchiang@mail.ncyu.edu.tw
*   Correspondence: wangxiaobo@ncut.edu.cn; Tel.: +86-010-8880-1541

**Abstract:** The environment can affect people's health by relieving stress, and rural landscape as a special environment might influence human's stress relief. This study takes different types of rural landscapes as the research object to explore their impact on stress levels, which are shown by photos. As an independent variable, the rural landscape is divided into three levels: Type 1 (natural landscape), type 2 (productive landscape), and type 3 (artificial landscape). Seventy-three subjects were randomly assigned to each type of rural landscape. Salivary cortisol, blood pressure, heart rate, and a subjective rating state scale (brief profile of mood states, BPOMS) were used as indicators of stress. At the same time, the influence of preference and familiarity on the stress relieving effect was also discussed. A paired *t*-test and one-way analysis of variance (ANOVA) were used as the main statistical methods. In the results of *t*-test for pre-posttest, significant difference was observed in high blood pressure, heart rate, and total mood disturbance (TMD) of type 1 and type 2, and the high and low blood pressure of type 3; ANOVA analysis revealed that for the difference of pre-posttest, significant difference was observed in the TMD value among the three types; except for type 3, blood pressure, heart rate, and BPOMS values were significantly affected by preference and familiarity. The conclusions include the following: The three types of rural landscapes have a positive effect on relieving stress; the productive landscape has the best effect on relieving stress; and users' landscape preferences and familiarity with the environment can affect the effect of stress relief in rural landscapes.

**Keywords:** rural landscape; stress; natural landscape; productive landscape; artificial landscape; preference; familiarity

## 1. Introduction

With increasing urbanization, rural landscapes are being involved in more and more research [1–3]. With the decline in the rural population, there have been phenomena such as hollow villages, changes in the appearance of the landscape, and urbanization of the rural areas in rural China. The service crowd, main functions, and morphological features of the rural landscape have undergone tremendous changes. The Chinese Beijing countryside, with its unique historical and cultural characteristics, is also affected by these problems. At the same time, the villages in Beijing are undergoing major renovations. In 2018 alone, there were 1081 villages in Beijing that completed the reconstruction plan, and about 60% of them built into "beautiful villages", which was a national policy of China. The future development of Beijing's rural landscape has become a worthy topic of investigation.

Meanwhile, many studies have proved that the environment could affect health, and the natural environment, in particular, has a positive effect on health, so it is considered to be a restorative environment [4–6]. Relieving stress may be one of the reasons that nature could promote human health. The theory of stress relief proposed by Roger Ulrich, from the perspective of evolutionary psychology, indicated that exposure to the natural environment could reduce stress [7]. Exposure to a real or simulated natural scene could reduce psychological and physiological stress by enhancing positive emotions and reducing negative emotions such as fear or anger [8]. Mitchell and Popham hypothesized that natural decompression effects explain the health benefits of having more green spaces nearby [9]. A previous study found that the most important motivations for visiting nearby green spaces were related to restoration [10].

Rural landscapes have more natural and green materials than cities. The rural landscape, with the characteristics of a restorative environment, has the potential to relieve stress. Research and theory on restorative environments provided a perspective on urban residents' desire for rural life, which was not only rural romanticism, but also evolutionary heritage [11]. People living in highly urbanized areas feel less healthy than those living in rural areas [12]. The rural environment reduces physiological stress, which is manifested by lowering the cortisol concentration, pulse rate, blood pressure, and a reported reduction in stress levels [3]. A review in rural settings found positive associations among pleasant aesthetics, trails, safety/crime, parks, and walkable destinations [13]. The health benefits of rural landscapes make it possible to expand the service and functions in the new era.

A number of studies have focused on the impacts of different landscape types on stress. Research has found that urban streetscapes are not as effective as urban parks, woodlands, and wild forests. [14]. Previous scholars explored the impact of four different naturalness landscape types, and show that exposure to the natural environment is more beneficial than the built environment on stress relief [15]. Some scholars have found that urban parks and urban forests could have a similarly positive impact for stress reduction, but the urban forest was a little better [16]. In the window views study, the green view has the best effect on students' stress recovery [17]. From these studies, we might infer that the more natural the environment is, the better stress relieving effect it could produce.

This paper studies the effects of different types of rural landscapes on stress relief, which was still rare in the previous research. This study can propose the types of rural landscapes worthy of being cherished from a health perspective, provide a reference for the renovation of the rural areas of Beijing, and help the public to choose the appropriate rural landscape to recover.

Kaplan's research showed that there was a complex relationship between preference, familiarity, and recovery [18], which has been less studied. Because the rural respondents were more familiar with grassland (a common landscape in their hometown), the rural respondents preferred grassland more than the urban respondents. [19]. This study investigates the preferences and the familiarity of the subjects, and attempts to analyze the relationship between preferences, familiarity, and stress relief. The results of the study help to provide personalized services to different people.

Based on the above analysis, this study divides the rural landscape into three types: Natural landscape, productive landscape, and artificial landscape according to the level of manual intervention. It analyzes the impact of the three types of rural landscapes on stress. Preference and familiarity were also included in a discussion on the impact of stress relief.

This present study proposes the following hypotheses: (1) Rural landscapes have positive effects on stress relief; (2) different types of rural landscapes have different effects on human stress recovery; (3) the more nature the type of rural landscape has, the more effective the stress reduction; (4) preference and familiarity would influence the effect of stress relief.

## 2. Materials and Methods

### 2.1. Material

Fifteen sample villages were chosen from more than 4000 villages in Beijing, China. The principle of rural selection referred to the five typical methods currently used for rural classification (Table 1). The sample villages could basically reflect the current rural landscape in Beijing. The selection method for 15 sample villages was as follows: First, the typical villages in a certain category were found from the relevant literature, government public information, etc. Second, those villages with more repetitions in each category were chosen. Third, the most representative villages were selected from the villages with more repetitions, and the sample villages were determined according to the operability of the field investigation. The locations of the sample villages are shown in Figure 1.

**Table 1.** Village samples in Beijing selection criteria.

| Serial Number | Classification Standard | Detailed Description | Village Name | Location in Beijing (District) |
|---|---|---|---|---|
| 1 | the distances of the villages from Beijing city | Village in the city | —— | —— |
| | | Village beside the city | Liu Li Qu Village | Men Tou Gou |
| | | Suburb village | Kang Ling Village | Chang Ping |
| | | Villages in outer suburbs | Peng He Yan Village | Mi Yun |
| | | | Mu Tian Yu Village | Huai Rou |
| | | | Bo Li Tai Village | Ping Gu |
| | | | Jie Shi Village, Cuan Di Xia Village | Men Tou Gou |
| | | Remote countryside | Gu Bei Water Town | Mi Yun |
| 2 | the appearance features of traditional rural landscape images | Artificial landscape dominated | Liu Li Qu Village, Cuan Di Xia Village | Men Tou Gou |
| | | Natural landscape dominated | Mu Tian Yu Village | Huai Rou |
| | | | Bo Li Tai Village | Ping Gu |
| | | Humanistic landscape dominated | Qian Jun Tai Village, Jiu Yuan Village | Men Tou Gou |
| | | | Kang Ling Village | Chang Ping |
| 3 | objects that the rural landscape has | Rural settlement landscape | Wu He Village | Fang Shan |
| | | | Sang Yu Village | Men Tou Gou |
| | | Rural productive landscape | Xiang Tun Village | Yan Qing |
| | | Natural ecological landscape | Jian Gou Village | Men Tou Gou |
| 4 | list of the most beautiful villages in Beijing | Urbanization consolidation | Bo Li Tai Village | Ping Gu |
| | | Reserved development type | Kang Ling Village | Chang Ping |
| | | Since 2006, an annual election "Finding the most beautiful village" is selected through a variety of ways, such as online voting, newspaper voting, and expert voting. | Kang Ling Village | Chang Ping |
| | | | Mu Tian Yu Village | Huai Rou |
| | | | Xiang Tun Village | Yan Qing |
| | | | Liu Li Tai Village | Ping Gu |
| | | | Ying Tao Gou, Cuan Di Xia Village, Jiu Yuan Village, Jian Gou Village | Men Tou Gou |

According to existing studies [15,20,21], environment was often divided into 4, 5, or 7 levels according to natural versus man-made conditions. Such as in one study, scholars, based on the levels of naturalness, divided the environment into four levels: Very natural, mostly natural, mostly built, and very built [15]. Considering the operability of the experiment, the classification of this study had been simplified. The rural landscape of Beijing villages was divided into three types based on the naturalness or intensity of human activities in the landscape (Table 2). The natural landscape (type 1) was dominated by naturally grown plants, terrain, and water, which could be perceived as real nature. The productive landscape (type 2) included farmland, vegetables, orchards, terraces, fences etc., which could be reminiscent of production. The artificial landscape (type 3) was full of villagers' houses, roads, walls, and so on, which could have the impression of gathering of human settlements in rural areas.

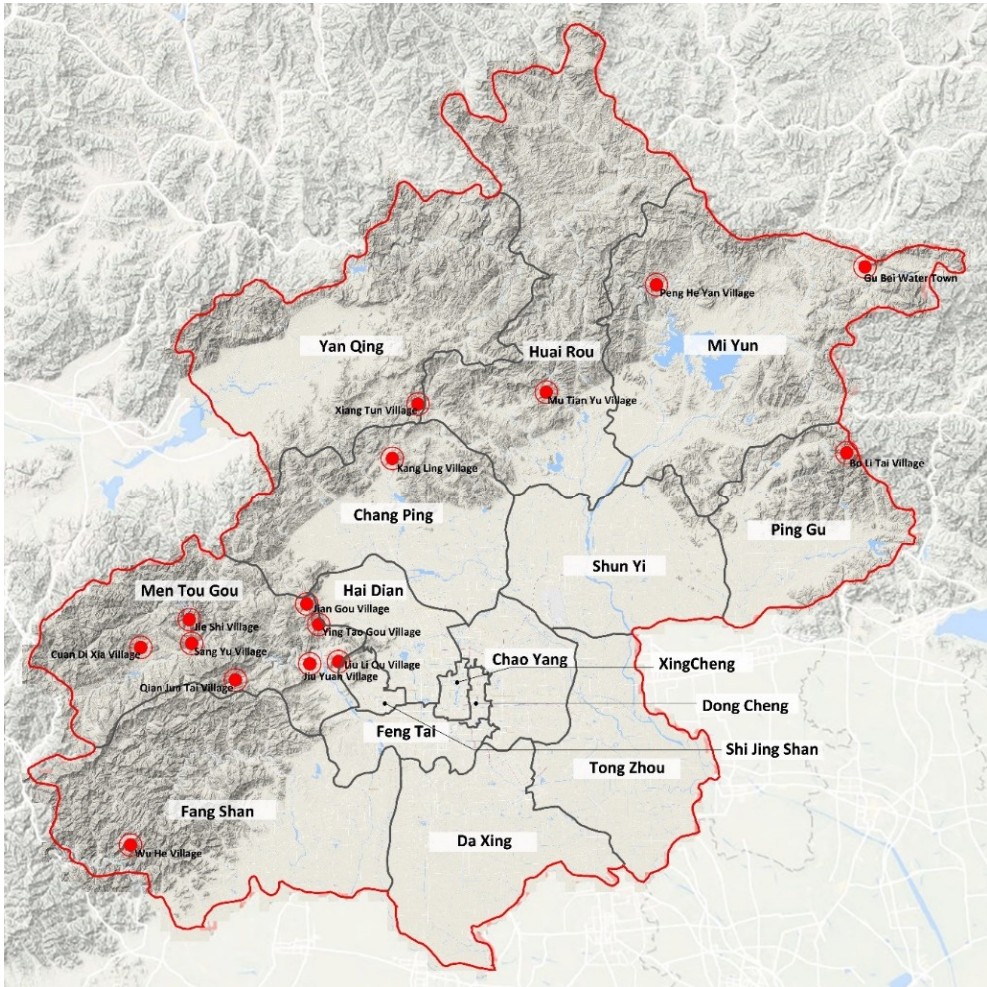

**Figure 1.** Location of the sample villages.

**Table 2.** Types of rural landscape selection criteria.

| Type | Naturalness | Artificial Degree | Detailed Description |
|------|-------------|-------------------|----------------------|
| 1 | Strong | Weak | Natural landscape |
| 2 | Relatively strong | Relatively weak | Productive Landscape (farmland, orchard, etc.) |
| 3 | Weak | Strong | Artificial Landscape (folk house, lane, etc.) |

The three types of rural landscapes were shown by photos in the laboratory in front of the subjects, because the sample villages were large in amount and far in distance. There were several previous studies that used photos or photos in slides instead of real scenes as the experiment materials, and they generalize well to onsite response [8,22–25]. Each type was represented by 25 photos, which were selected from more than 1600 original photographs across the 15 sample villages. The principle purpose of photo selection was to fully reflect the main features of this type of rural landscape. Type 1 photos were mainly natural environments in the countryside where there was no or little human intervention, containing plants, terrain, bodies of water, etc. Type 2 photos were production environments with farmland, orchard, fence, terrace, etc. Type 3 photos were artificial landscapes including residential houses, country roads, squares, etc. There were green plants in each type of rural landscape. The green quantity from high to low was type 1, type 2, and type 3. The photos were taken from a human point of view to get close to the experience of the scene. The photos contained a variety of landscape elements without people and animals. Photos were taken in the same season during the day from April to June

2016 in similar weather (sunny), with a Canon EOS 60D, and they were clear, had the same resolution, and were uniform (Figure 2).

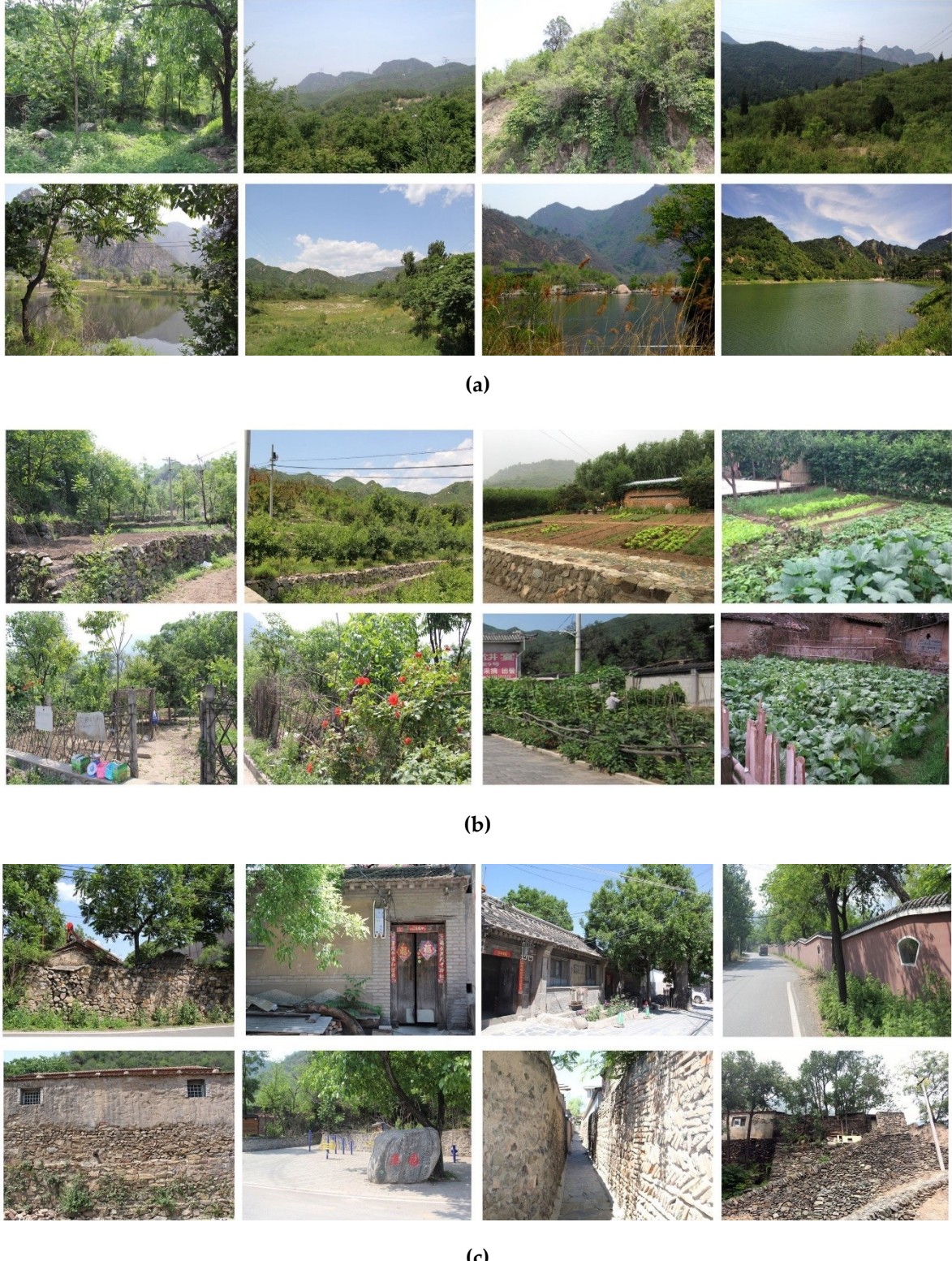

**Figure 2.** Types of rural landscape. (**a**) Type 1 natural landscape; (**b**) type 2 productive landscape; (**c**) type 3 artificial landscape.

## 2.2. Subjects

Most of the subjects were college students, who were physically healthy and had no mental or salivary gland diseases. The 73 valid subjects were chosen from 76 candidates, where 3 persons did not fulfil the requirements or the data were not complete. Most of these students were from North China University of Technology. Most of them were from Beijing, from urban to rural. They were randomly selected and had no specific majors. The time at the university was 1–6 years, with an average age of 22.84 ± 2.03. There were 51 females in the test, accounting for 69.9%, and 22 males, accounting for 30.1%. Seventy-three subjects participated in the blood pressure, heart rate, and brief profile of mood states (BPOMS) scales. Due to limited funding, only 44 subjects participated in the salivary cortisol test.

In recruiting the subjects, besides the normal demographic issues, the preferences for rural landscape types, rural life experiences, and professional information of the subjects were also collected to reflect the subjects' preferences and familiarity with the rural landscape.

All subjects gave their informed consent for inclusion before they participated in the study. The study was conducted in accordance with the Declaration of Helsinki, and the protocol was approved by the Ethics Committee of North China University of Technology (NCUT).

## 2.3. Experimental Design

This study adopted an independent group design.

The independent variable was the rural landscape, which was divided into three categories: Type 1, type 2, and type 3 (Table 2). The dependent variable was the stress state. Stress status was assessed through a series of physiological and psychological indicators. Physiological indicators included salivary cortisol, blood pressure, and heart rate; psychological indicators were the brief profile of mood states (BPOMS), and the total mood disturbance value (TMD) in the scale was an important indicator of emotional state [22].

Subjects were randomly assigned to the three types of rural landscapes, and each subject participated in only one independent variable level test. The test was performed independently; that is, only one test was accepted at a time to avoid cross interference. The assignment of the subjects, including the salivary cortisol (sCort) test and other experiments, are shown in Figure 3.

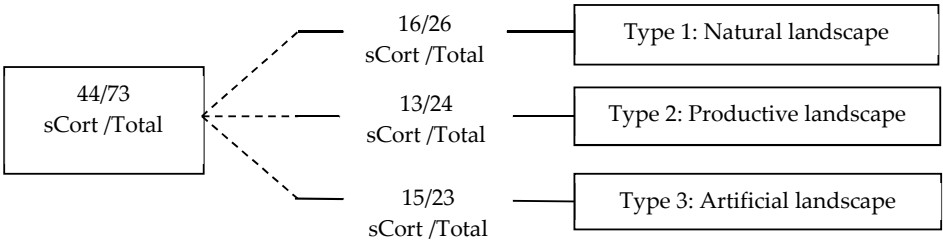

**Figure 3.** Independent group design.

## 2.4. Experimental Procedure

There were three steps and three measurements during the experiments (Figure 4).

First, when a subject arrived at the study laboratory, he/she was given a brief introduction about the experiment and was asked to turn off his/her cell phone and not use any electronic media for the remainder of the experiments. Then, his/her blood pressure and heart rate were measured for the first time, the aim of which was to collect the basic physiological data and make the subject familiar with the measuring instruments (Time 1, T1).

Second, the subject was asked to do a Trier social stress test (TSST) [23], which is widely used as a classic psychosocial stress paradigm conducted in a laboratory setting. The subject was asked to give a five-minute speech in public and do subtraction problems for the purpose of raising the stress levels. Then, the subject provided a pre-exposure saliva sample, the blood pressure and heart rate were measured again, and the subject was asked to do the BPOMS scale (Time 2, T2).

Third, the subject was randomly assigned to see photos of one of the three types of rural landscapes. The subject was told not only to watch the photos but also to imagine the feeling of being there. Each photo was shown for 15 s, followed by 3 s of a blank screen before the next image in the experiment, according to previous research [24–30]. After this, post-exposure salivary, blood pressure, heart rate, and BPOMS scale data were collected (Time 3, T3).

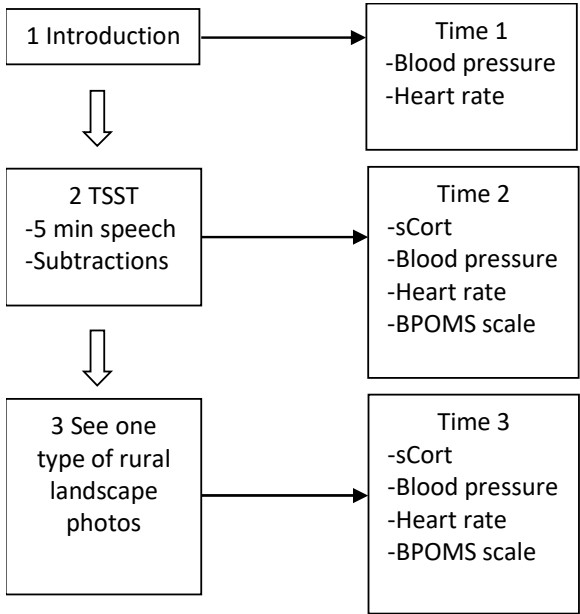

**Figure 4.** Procedure.

*2.5. Measurements*

Stress levels were reflected by physiological and psychological indicators.

2.5.1. Physiological Index

● Salivary cortisol (sCort)

Salivary cortisol concentrations can reflect human stress levels. Because of its simple operation and objective quantification, it is often used in experimental research [3,15,16,31–33].

In this study, salivary cortisol was collected before and after seeing the rural landscape photos at Time 2 and Time 3 (T2 and T3). Each saliva sample was collected with a professional cotton ball and stored in a matching test tube. Each cotton ball was spit into a test tube after 45 s of chewing. Each tube was numbered for later analysis, and the tube was stored in a 2–5 °C refrigerator. A salivary cortisol ELISA, SLV-2930 (DRG Instruments GmbH, Germany), was used as the reagent kit, which is an enzyme immunoassay for the quantitative in vitro diagnostic measurement of active free cortisol (hydrocortisone and hydroxy corticosterone) in saliva.

● Blood pressure and heart rate

Blood pressure and heart rate can also reflect people's stress level [4,33,34], especially the changes in data before and after the experiment.

In this study, OMRON electronic sphygmomanometer and upper arm HEM-7111 were used to measure blood pressure and heart rate. Blood pressure and heart rate were collected at Time 1, Time 2, and Time 3 of the three steps.

2.5.2. Psychological Index

● Brief profile of mood states (BPOMS)

The profile of mood states (POMS), a validated 65-item scale, is an effective tool to research mood state that was first compiled by McNair in 1971 [35].

Simplification, revision, and testing were performed to make a brief profile of mood states (BPOMS) consisting of 30 items [22,30], each using a five-point Likert scale format [31,32,36,37]. The scale includes five dimensions, namely, tension (T), anxiety (A), fatigue (F), vigor (V), and confusion–depression (C+D). By subtracting the vigor score from the sum of all the other mood scale scores, an assessment of total mood disturbance (TMD) was calculated (1). The TMD was considered a highly reliable and clinically relevant measure of overall mood and mood problems, with higher scores indicating greater mood disturbance.

$$TMD = T + A + F + C + D - V \tag{1}$$

A high TMD score indicates a poor emotional state. Except for V value, the higher the dimension (including T, A, F, C + D) values, the worse the mood is.

### 2.5.3. Landscape Preference and Familiarity

Information on landscape preferences and familiarity was collected by questionnaire. Subjects were asked to choose one type that they liked from the three types of rural landscapes. Familiarity was reflected by four questionnaire questions: (1) The longest time spent in your life is in the city or the country; (2) whether you have a rural life experience; (3) the length of your rural life experience—about one week, about one month, about one quarter, about half a year, one year or more; (4) whether your major is landscape architecture or related majors, including architecture, urban, and rural planning. If the subject had a rural life experience, long experience with rural life, and studied landscape architecture and related majors, he/she would be considered to be more familiar with rural landscapes.

### 2.6. Statistical Analysis

This study used SPSS 20.0 and Excel 2010 software for data analysis. Each indicator that reflects the stress level followed these steps: First, it was verified by Q-Q diagram analysis whether all the experimental data satisfied the normal distribution. Second, the homogeneity of variance test was done, which is shown in Appendix A. Third, outliers were found through box-plot, which were removed from the subsequent statistical analysis. Then, different patterns of tests were used to differentiate data. A paired *t*-test was used to determine whether there was any difference between the pre-test (Time 2) and post-test (Time 3); that is, whether the rural landscape had played a role in stress restoration. One-way analysis of variance (ANOVA) was used to test the differences between the pre-test (Time 2) and the post-test (Time 3) (T2–T3), which was aimed to find whether different types of rural landscapes had different influences on stress reduction. In addition, the average difference between the post-test (Time 3) and the pre-test (Time 2) (T2–T3) was compared to reflect the impact of different rural types. Finally, one-way analysis of variance (ANOVA) was used to analyze the physiological and psychological index with preference and familiarity, which was aimed to reflect the relation among stress recovery, preference and familiarity.

## 3. Results

The paired *t*-test showed that for the physiological index, there was no significant difference in the pre-post measure of salivary cortisol concentration. But significant difference was observed in the pre–post measure of high blood pressure and heart rate in rural landscape type 1 and type 2, and the high and low blood pressure of type 3. For the psychological index, significant difference was observed in the pre–post measure of TMD for type 1 and type 2.

ANOVA analysis of the mean difference (T2–T3) in physiological index showed no difference among the types, but in the psychological index, revealed significant differences. Multiple comparisons indicated that type 2 was obviously significant different from type 1 and type 3, especially between type 2 and type 3 ($p < 0.01$).

### 3.1. Salivary Cortisol

A box-plot was made for pre-test (T2) and post-test (T3) of salivary cortisol, as shown in Figure 5. Two outliers were removed from the subsequent statistical analysis (Figure 5).

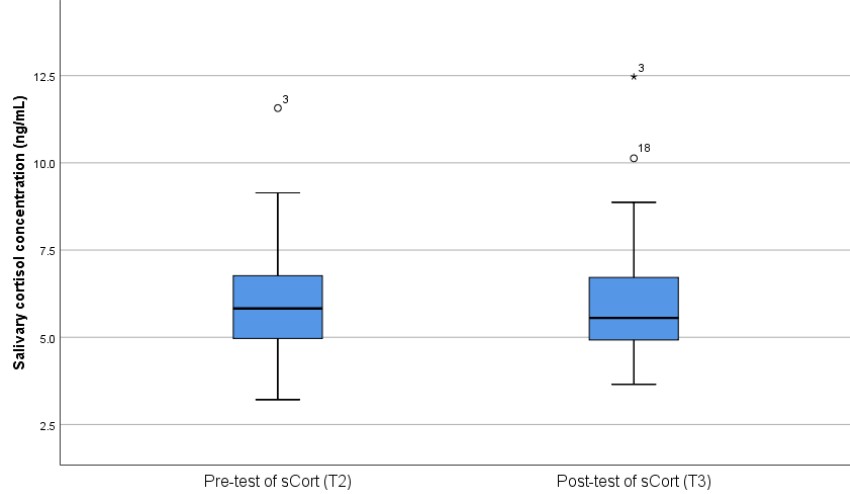

**Figure 5.** Box-plot of pre-test and post-test of salivary cortisol. In this chart, two outliers were found, which were No. 3 and 18.

### 3.1.1. The Paired *t*-test and ANOVA Analysis

The relevant sample *t*-tests of the pre-post measure for different types of rural landscapes were not significant. Type 1, t (14) = −1.333, *p* = 0.204; type 2, t (11) = 1.573, *p* = 0.144; type 3, t (14) = 0.464, *p* = 0.650. One-way ANOVA was performed on the difference (T2–T3) in salivary cortisol concentrations in different types. The results were not significant: F (2,39) = 2.399, *p* = 0.104.

### 3.1.2. The Average Trend

The average data showed that the post-test values were both smaller than the pre-test in type 2 and type 3, but adverse in type 1. The average value of the difference (T2–T3) in the salivary cortisol concentration was compared, and type 2 was the highest, which might indicate that of the three types of rural landscapes, the salivary cortisol concentration of type 2 decreased the most and the decompression effect was the best (Figure 6).

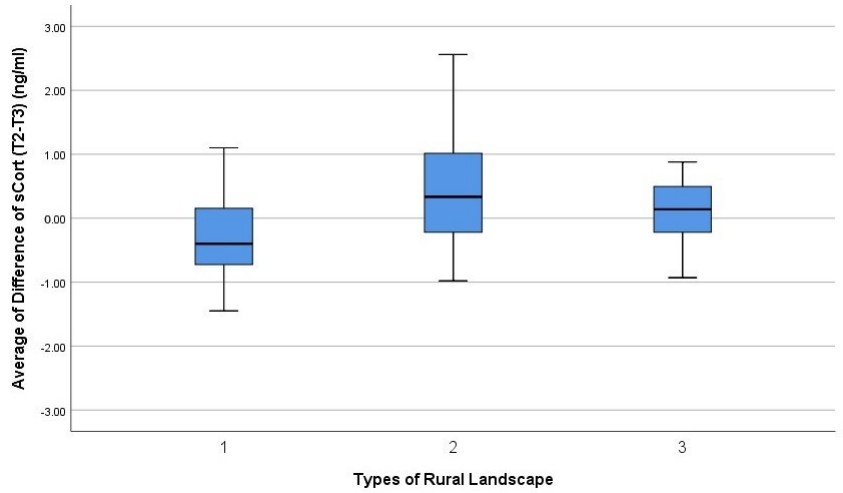

**Figure 6.** Box-plot of the average of the difference in salivary cortisol concentration (T2–T3).

### 3.2. Blood Pressure, Heart Rate

A box-plot was made for pre-test (T2) and post-test (T3) of blood pressure and heart rate, as shown in Figure 7. Five outliers were removed from the subsequent statistical analysis (Figure 7).

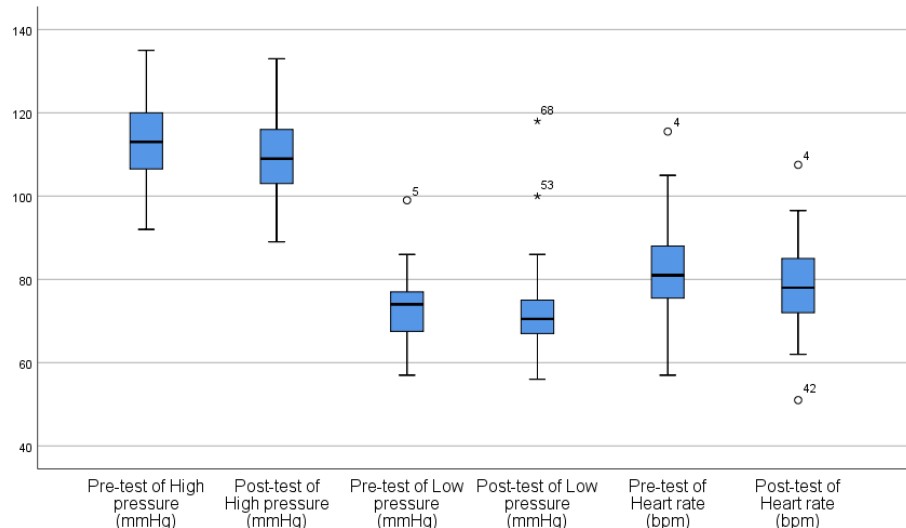

**Figure 7.** Box-plot of pre-test and post-test of blood pressure and heart rate. In this chart, five outliers were found, which were No. 4, 5, 42, 53, and 68.

### 3.2.1. The Paired *t*-test

The relevant sample *t*-test of the pre–post measure of different types of rural landscapes showed that there were significant different in all types. Type 1, high pressure t (23) = 2.19, *p* = 0.04; heart rate t (23) = 2.14, *p* = 0.04. Type 2, high pressure t (22) = 3.13, *p* = 0.01; heart rate t (22) = 2.14, *p* = 0.04. Type 3, high pressure t (20) = 2.87, *p* = 0.01; low pressure t (20) = 2.59, *p* = 0.02.

### 3.2.2. ANOVA Analysis

One-way ANOVA was performed on the difference (T2–T3) in high and low pressure and heart rate in different types of rural landscapes. The results were not significant. High pressure, F (2,65) = 0.352, *p* = 0.704; low pressure, F (2,65) = 0.712, *p* = 0.494; heart rate, F (2,65) = 0.001, *p* = 0.999. From the analysis of the data, it could be seen that there was no statistical difference in blood pressure and heart rate among the three types of rural landscapes.

### 3.2.3. The Average Trend

The average data showed that the post-test values were all smaller than the pre-test, which indicated that blood pressure and heart rate all dropped through seeing the rural landscapes.

### 3.3. BPOMS Scale

A box plot was made for pre-test (T2) and post-test (T3) of TMD, as shown in Figure 8. Five outliers were removed from the subsequent statistical analysis (Figure 8).

### 3.3.1. The Paired *t*-test

The relevant sample T-test for the pre-post measure of different types suggested that there were significant differences in the TMD values and each dimension value, including: Type 1, TMD t (24) = 2.49, *p* = 0.02, T t (24) = 5.89, *p* = 0.00, and C + D t (24) = 2.48, *p* = 0.02; type 2,TMD t (22) = 4.31, *p* = 0.00, T t (22) = 3.324, *p* = 0.00, F t(22) = 2.18, *p* = 0.04, V t (22) = −2.89, *p* = 0.01, and C + D t (22) = 4.11, *p* = 0.00; type 3, T t (19) = 3.68, *p* = 0.00, and V t (19) = 2.46, *p* = 0.02.

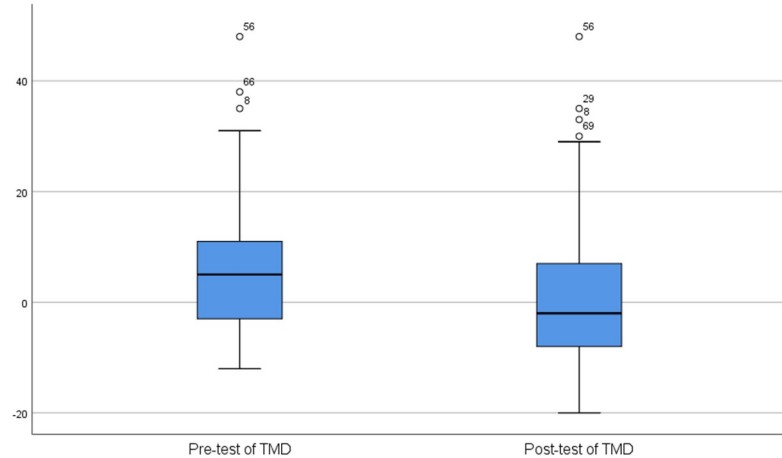

**Figure 8.** Box-plot of pre-test and post-test of total mood disturbance (TMD). In this chart, five outliers were found, which were No. 8, 29, 56, 66 and 69.

### 3.3.2. ANOVA Analysis

One-way ANOVA was performed on the difference (T2–T3) in BPOMS in different types of rural landscapes. The TMD and V values were statistically different. Multiple comparisons showed a significant difference between type 1 and type 2, and type 2 and type 3 (Tables 3 and 4).

**Table 3.** ANOVA analyses of brief profile of mood states (BPOMS) values with statistical difference.

|  |  | SS | df | MS | F | Sig. |
|---|---|---|---|---|---|---|
| Difference of TMD | SSA | 666.72 | 2 | 333.36 | 4.54 ** | 0.01 |
|  | SSE | 4768.14 | 65 | 73.36 |  |  |
|  | SST | 5434.87 | 67 |  |  |  |
| Difference of V | SSA | 212.55 | 2 | 106.28 | 6.04 ** | 0.00 |
|  | SSE | 1142.92 | 65 | 17.58 |  |  |
|  | SST | 1355.47 | 67 |  |  |  |

** $p < 0.01$.

**Table 4.** Multiple comparisons of TMD and vigor (V) among the three types.

| LSD | (I) | (J) | Mean Difference (I-J) | Sig. |
|---|---|---|---|---|
| Difference of TMD | Type1 | Type2 | −5.09 * | 0.04 |
|  |  | Type3 | 2.58 | 0.32 |
|  | Type2 | Type1 | 5.09 | 0.04 |
|  |  | Type3 | 7.67 ** | 0.01 |
|  | Type3 | Type1 | −2.58 | 0.32 |
|  |  | Type2 | −7.67 ** | 0.01 |
| Difference of V | Type1 | Type2 | −2.00 | 0.10 |
|  |  | Type3 | 2.46 | 0.06 |
|  | Type2 | Type1 | 2.00 | 0.10 |
|  |  | Type3 | 4.46 ** | 0.00 |
|  | Type3 | Type1 | −2.46 | 0.06 |
|  |  | Type2 | −4.46 ** | 0.00 |

* $p < 0.05$, ** $p < 0.01$.

### 3.3.3. The Average Trend

The average data showed that the post-test values were almost smaller than the pre-test in every subscale (TMD, T, A, F, C + D), which indicated that the mood was better after seeing the rural landscapes. However, the V value was a little complicated. In type 1 and type 2, the post-test V value was larger than the pre-test, but smaller in type 3. This implied that type 1 and type 2 made vigor increase, but type 3 made vigor less. The figures of BPOMS including TMD, T, A, F, V, and C+D are shown in Table 5. The pre-test was T2, and the post-test was T3.

**Table 5.** The figures of BPOMS.

| Types | TMD | | T | | A | | F | | V | | C+D | |
|---|---|---|---|---|---|---|---|---|---|---|---|---|
| | T2 | T3 | T2 | T3 | T2 | T3 | T2 | T3 | T2 | T3 | T2 | T3 |
| 1 | 4.48 | 0.40 | 2.36 | 0.44 | 1.24 | 0.80 | 2.64 | 2.80 | 7.28 | 8.24 | 5.52 | 4.20 |
| 2 | 3.52 | −5.65 | 1.96 | 0.61 | 1.04 | 0.74 | 3.00 | 1.52 | 8.74 | 11.70 | 6.26 | 3.17 |
| 3 | 4.40 | 2.90 | 2.10 | 0.60 | 0.95 | 0.70 | 3.60 | 3.85 | 9.05 | 7.55 | 6.80 | 5.30 |

Type 2 had the highest average value of the difference (T2–T3) in TMD. It indicated that in the three types of rural landscapes, type 2 had the best effect on improving emotional state (Figure 9).

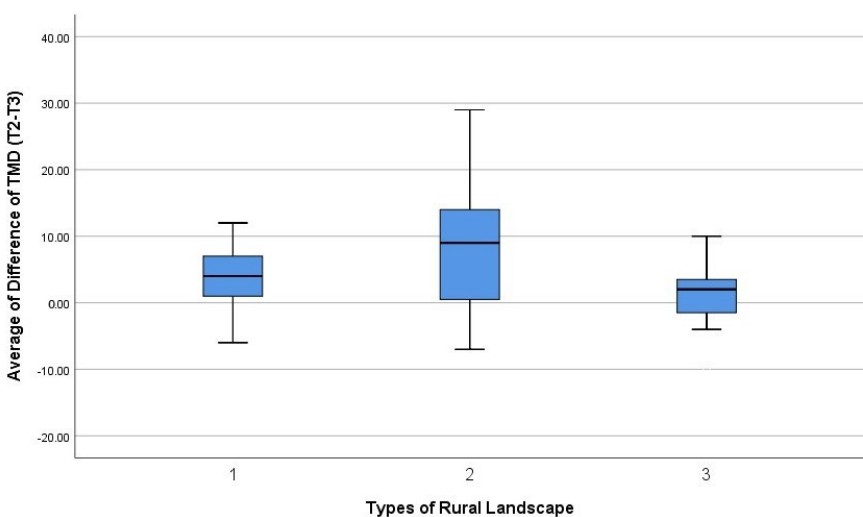

**Figure 9.** Box-plot of the average of the difference in the TMD (T2–T3).

### 3.4. Various Indicators of Landscape Preference and Familiarity

In the experiment, the relationship between stress relief, preferences, and familiarity was analyzed.

- The salivary cortisol concentration had no significant difference among these factors (the landscape preferences, rural life experience time, and professional aspects).
- The difference in blood pressure was affected by the length of life in the village. There was a significant difference between the difference in high pressure (T2–T3), and when the time spent in rural life was one week and one month ($p = 0.043$), one month, and one year or more ($p = 0.046$). For low pressure, the preference was better; that is to say, the subject watching the type of rural landscape that they liked showed a greater reduction in low pressure; except for type 3, the other two types' high pressure was the same as the low pressure.
- The difference in heart rate was also affected by the length of life in the village. There was significance for 0 and one month ($p = 0.037$), and one month and one year or more ($p = 0.019$). Except for type 3, the preferred landscape could reduce the heart rate more.

- Except for type 3, the pre-post difference in the TMD value in the BPOMS, the preferred type could reduce the TMD value more.

## 4. Discussion

### 4.1. The Three Types of Rural Landscapes Have a Positive Effect on Stress Relief

The three types of rural landscapes have a positive effect on the relief of stress. The data of the pre–post measure, which were analyzed by the relevant sample *t*-test, proved that there was a statistically significant difference between the blood pressure, heart rate, and BPOMS values in all types of rural landscapes. Blood pressure, heart rate, and BPOMS value all decreased. This suggested that all the three types of rural landscapes could reduce stress levels to a certain extent, both physically and psychologically. The results of this study confirmed previous research [10,11]. This might be related to the greenery of the three types of rural landscapes, which was consistent with previous research findings in the green environment to relieve stress [7,9] and contact with nature to promote health.

For the natural landscape and the productive landscape, in the physiological index, significant difference was found in the blood pressure and heart rate; in the psychological aspect, there was significant difference in the TMD value of BPOMS scale.

For artificial landscape, the high pressure and low pressure significantly reduced. It indicated that the artificial landscape could relieve stress in physiological aspect. But for the psychological aspect, the artificial landscape was contradictory. For tension (T) value, the post-test was smaller than the pre-test, which implied that the mental tension was alleviated, and this was positive for stress relief. Meanwhile, in vigor (V) value, the post-test was smaller than the pre-test, which implied that the mental vigor was reduced, and this was negative for stress relief. This research perspective could also be found in past studies [38].

### 4.2. There are Psychological Differences in the Relief of Stress among Different Types of Rural Landscapes

Different types of rural landscapes have different effects on stress levels. There was no significant difference in physiological indexes among the three types of rural landscapes; but psychologically, the mean difference in the pre-post measure (T2–T3) of the TMD was significant different. Further multiple comparisons showed the difference between the natural landscape and the productive landscape was significant ($p < 0.05$), and the difference between the productive landscape and the artificial landscape was more significant ($p < 0.01$).

This might imply that there was no significant difference in the physiological stress relief among the three types of rural landscapes, but the differences between the natural landscape and the productive landscape, and between the productive landscape and the artificial landscape were very significant from the perspective of people's subjective psychological feelings. This kind of stress-relieving effect was not significantly different in terms of physiology, but obvious psychological differences have appeared in a previous comparative study of urban historical blocks and forest environment by Ulrika Stigsdotter et al. [39]. This finding could be confirmed by previous research that green spaces were associated with better self-perceived general health and better mental health [40].

For the effect of stress relief, the natural landscape was not as obvious as the productive landscape. But for some physiological and psychological indicators, it also had a good stress recovery effect. The effect was second only to the productive landscape, and higher than the artificial landscape.

### 4.3. The More Natural, not Necessarily the Better: The Decompression Effect

The results of the study showed that in the three types of rural landscapes, the natural landscape was not the one that reduced stress the most. The best one was the productive landscape. Except for the change in blood pressure, the other values of the productive landscape's indicators were ranked first. Multiple comparisons indicated that in the TMD value, the natural landscape and the artificial landscape did not have significance difference, but both of them had significance difference from the

productive landscape. In the V value, the productive landscape and the artificial landscape had a significance difference. This result was different from some existing research conclusions. Some studies suggested that for stress reduction, more nature was better [14–17,41,42]. However, some studies have not indicated that the more natural the environment, the better the effect of relieving stress. As Kaplans (1989) pointed out, more natural environment is not necessarily the environment with the best preference [43]. Gobster discovered that the spatial configurations of people's preferences were less ecological [44]. One study found that no correlation between biodiversity (based on number of plant species) and the duration of real time spent by the citizens at the green spaces was found [45]. These studies confirmed the findings of this present study.

The productive landscape had a good performance in the three types of rural landscapes for stress relief. The possible psychological basis was that type 2 as a productive landscape, such as farmland and orchard, could bring food to people and meet basic survival needs. People had a sense of security and satisfaction when they saw the productive landscape, which might reduce stress. The idyllic landscape with a productive function is a type of landscape that is worthy of cherishing. The future village development should pay attention to the protection of this type of rural landscape.

### 4.4. The Effects of Preference and Familiarity on Stress Relief

The present study results on the stress relieving effects and the landscape preference and familiarity of the subjects in this study are worth discussing as follows: (1) Regarding preference, except for the artificial landscape, the greater was the preference, the better the stress relief effect. (2) Regarding familiarity, the more familiar, the better was the stress relief effect. For rural life experience, one month and one year is a time limit with demarcation significance. For subjective TMD values, subjects in the relevant professions were better at relieving stress.

This suggests that we should fully understand the preferences of users in the landscape environment because personal characteristics will affect preferences; this is consistent with existing research [46]. And a previous study found that the preference for environments was related with the state of fatigue, the more fatigue the more prefer for nature [47]. This prompts us to fully understand the preferences of users in the landscape environment because different landscape types may have different recovery stress effects for different users. At the same time, users need to fully understand the environment; the more understanding, the better is the subjective effect on stress relief. These findings were supported for specifying who might benefit most and least from the relationship between nature and health happens, which was mentioned in the previous study [38]. For the public, it is necessary to strengthen the related publicity and education of rural landscapes and thus promote the diversified and healthy development of the countryside.

### 4.5. Limitations and Future Research

The method for determining the concentration of salivary cortisol needs to be improved. The index of the salivary cortisol concentration did not show a significant decrease in the pre–post-test of this study, but it increased in the natural landscape; this was inconsistent with previous studies [15,31]. It might be caused by both time control and the small sample size. Research showed that the subjects who experienced the Trier social stress test (TSST) objectively showed significant stress physiological responses, and the cortisol content in saliva increased, but the peak of salivary cortisol concentration generally appeared 10 min after the test [48]. There was hysteresis. In this experiment, the measurement time of the salivary cortisol concentration was not precisely controlled, and the premeasured salivary cortisol concentration (T2) did not clearly indicate the highest value of the salivary cortisol concentration after the TSST situation; the post-measured salivary cortisol concentration also did not clearly show the value of salivary cortisol concentration after seeing the rural landscape. At the same time, the sample size was small. Due to the limitation of funds and time, this study only tested the salivary cortisol concentration for 44 subjects (type 1, 16; type 2, 13; type 3, 15). The sample size was smaller than other indicators.

Photographs were used in this study, which reduced the ecological validity of the study. Photos could show many rural landscape scenes at the same time and could be presented to the subjects in a simple way, which made the experiment easy to operate. However, watching photos was different from the real environment, which had a more realistic experience. This was essentially a still and two-dimensional experience, which does not reflect the dynamic and four-dimensional experience of a real landscape. This has similar limitations to other studies using photographs as research materials [23–25]. Perhaps watching Virtual Reality (VR) films is a better research method that could both have operability and real experience. Studies manipulating real-world settings would have more ecological validity.

The three types of environments in this study belong to the same rural landscape and have some inherent consistency. The results of this study show that the artificial rural landscape (including artificial buildings such as rural buildings and roads) also has a certain relief effect on stress. The environment of the productive rural landscape is better for stress relieving than the natural landscape. Previous research has studied different types of environment, such as cultivated fields, forest, golf, commercial landscapes, industrial areas, brick building wall, etc. [5,42,49]. Most of the findings suggested that nature provides greater positive health effects compared to urban views, but comparisons between subcategories of nature showed fewer clear results in health effects [50], just like the present study. More in-depth study of the different types of landscape would be a good direction for future research.

In the study by Ulrika Stigsdotter et al., the research content was also inherently consistent [39], just like the present study. This consistency was that the research objects were all of good quality, both urban historic neighborhoods, and forest environments. The choice of such research objects has led to the breakthrough of previous research in the field of stress relief that nature is superior to the urban environment, or the more natural, the greater is the stress relief effect.

These research conclusions suggest that, just from the perspective of types, it is difficult to find significant differences among the environments with the same characteristics. This may also mean that in addition to the types, certain factors (such as aesthetics, culture, form, space, scale, material, etc.) contained in the environment may be more essential factors affecting stress recovery. Research on environmental "traits" already exists, including the four elements of the restorative environment proposed by Kaplan [43] (being away, compatibility, extent, and fascination), and the perceived sensory dimensions (social, prospect, rich in species, serene, culture, nature, and refuge) [51,52]. These studies will give us a clearer understanding of the laws of the environment on stress reduction, which is worth exploring in future research.

The scenic beauty estimation (SBE) method is a system for quantitatively indexing the aesthetic quality of landscapes. The most frequent applications of the SBE method were to report the judgments of photos in terms of a 10-point rating scale [53], which was mainly relying on subjective evaluation. In the present study, in addition to subjective evaluation (BPOMS), we also applied some objective physiological indicators, such as salivary cortisol, blood pressure, and heart rate. More objective physiological indicators should be applied in future research. A previous study about SBE found the highest correlation of the landscape metrics-based assessment with the visual assessment results of the photographs [54]. This study might suggest a correlation between SBE and naturalness. As further inferences, there might be some correlation between SBE and stress recovery, which is worth exploring in future research.

## 5. Conclusions

Rural landscapes have a positive effect on relieving stress, which is particularly prominent at the reduction of blood pressure, heart rate, and the level of psychological perception.

Three types of rural landscapes have different effects on stress relief, but it is not true that the more natural the rural landscape is, the better the stress relief effect. The productive landscape has the best effect at relieving stress.

Different types of rural landscapes have no differences in physiology for stress relief, but there are differences in psychology. Psychologically, the level of stress relief felt by the productive landscape is significantly better than the other two types of rural landscapes—the primitive natural and the artificial landscapes.

The user's landscape preferences and familiarity with the environment can affect the effect of stress relief in rural landscapes. For the type of rural landscape dominated by natural features, the more people like the type, the better the effect of relieving stress. When choosing a natural environment for stress recovery, it is necessary to first determine the type that the user likes to achieve better results. The more familiar people are with the rural landscape, the better is the effect of relieving stress. In addition to the familiarity of their own experience, the more people understand the rural landscape, the better is the effect of relieving stress.

**Author Contributions:** Writing—original draft preparation, X.W. and H.Z.; writing—review and editing, X.W. and H.Z.; supervision, Z.S. and Y.C.; funding acquisition, X.W.

**Funding:** This research was funded by the National Natural Science Foundation of China, grant number 51708003.

**Acknowledgments:** Thanks to Yaxing Shi and Liang Zhang for a lot of first-hand experimental operation work.

**Conflicts of Interest:** The authors declare no conflicts of interest.

## Appendix A

Through Q-Q graph analysis, each group of data approximated a normal distribution. From the significance of the variation number isomorphism test, $p > 0.05$, indicating the homogeneity of variance. Due to the limitation of the layout, only the salivary cortisol difference (T2–T3) with the smallest sample size were presented on Q-Q diagram analysis for normality and test of homogeneity. (Figure A1, Table A1).

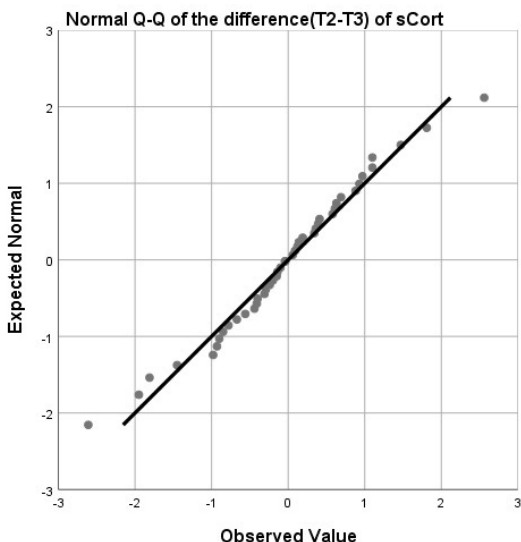

**Figure A1.** The Q-Q diagram analysis for normality.

**Table A1.** Test of homogeneity of variances of the difference of salivary cortisol concentration.

| Levene Statistic | df1 | df2 | Sig. |
|---|---|---|---|
| 0.849 | 2 | 41 | 0.435 |

$p > 0.05.$

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
