# Peer review of "The Influence of Viewing Photos of Different Types of Rural Landscapes on Stress in Beijing"

_sustainability, doi:10.3390/su11092537_

Round 1

Reviewer 1 Report

Authors improved the paper, which I still consider as Very interesting, valuable and well-designed, because of its interdisciplinary view on cultural landscape.

Regarding abstract authors added the connection between rural landscapes and experiment with human subject, making it better understood. Introduction was a bit adjusted including more precise wording and shortened, but I still dare say, that is too long.

Corrections made on the text, especially explanation for dividing naturalness into three-level, makes section of methods better, but I do not understand well what is criteria of assigning landscapes into that three levels and name it “very” and “mostly”. What is the border between “very” and “mostly”, did You use any number-parameters?  

I am sorry making you puzzled but Scenic Beauty Estimation is still showing people pictures of the landscape and expecting answer of valorization, the difference is – in SBE people just mark pictures by giving them points, in Your method reaction is in blood pressure, hart rate, cortisol level etc. That I think method SBE should be discussed (e.g. subsection 4.5). But discussion is really interesting and dividing it into proper section makes argument easy to understood.

I am not the expert of English language, but changes of some words made it easier to read, that I believe language was improved.

Author Response

We appreciate your affirmation and encouragement again. We are very grateful for your valuable comments and take all of them seriously. We feel very strongly that our manuscript is greatly improved as a result of your helpful comments.

We hope that our revised manuscript satisfactorily addresses all issues and that it is now suitable for publication.

Below are our responses to your comments. The page and line numbers please refer to our revised manuscript.

We adjusted the introduction to make it more concise again, please see L36-37, L43, L51, L60-61, L63-64, L69-70, L72, L74-75.

Scenic Beauty Estimation (SBE) is such an important method, so we discussed it in two aspects in subsection 4.5: 1) the most frequent applications of the SBE Method were more subjective, well this present study involved some objective physiological indicators. And More objective physiological indicators should be applied in future research; 2) A previous study suggest a correlation between SBE and naturalness. Well, the correlation between SBE and stress recovery is worth exploring in future research. Please see L477-486. Thank you so much for your mentioning SBE. We think it’s an excellent idea and an important addition to our manuscript.  

For dividing naturalness into three-level, we didn’t use number-parameters, that should be improved in our future research. We referred the classification of Kurt Beil and Douglas Hanes, which was not accurate too. But it has some representativeness and could explain some problems. Just like many studies in the field of psychology. Well, quantitative research is very important and more precise. We will use more quantitative methods in the future research. Thanks a lot for your suggestion.

Reviewer 2 Report

Thanks for all the changes - it is a big improvement and a lot more clear. However, you have not adequately addressed my issue with the fact that it is photographs. The few lines in the discussion are not enough to explain this so I still wish to see something in the methods which explains how photos can be seen as adequate surrogates for real landscapes - there is plenty on this to select for the review - and to justify how this can satisfy the experimental design. It needs to be clearly stated earlier in the work that it is based on photographs - probably in the title and abstract as well. The limitations are not really strongly enough recognised either.

Author Response

We appreciate your affirmation and encouragement again. We are very grateful for your valuable comments and take all of them seriously. We feel very strongly that our manuscript is greatly improved as a result of your helpful comments.

We hope that our revised manuscript satisfactorily addresses all issues and that it is now suitable for publication.

Below are our responses to your comments. The page and line numbers please refer to our revised manuscript.

In the materials and methods part, we added the use of photos to reflect the real environment and found more literature references to support this approach, please see L123-125. We also discussed more of the limitations for the use of photos and added some references, please see L447-451. We hope that these additions, especially the addition of references, would provide a clearer explanation of the reasons and limitations of the using photos in the experiment.

We emphasized in the title and abstract that the research is based on photos, please see L2, L16-17.

By the way, the using of VR film is such a good idea. Actually, we have already adjusted our current research by using this method. We feel it’s a major improvement in our research. Thank you so much for that again.

This manuscript is a resubmission of an earlier submission. The following is a list of the peer review reports and author responses from that submission.

Round 1

Reviewer 1 Report

This is a paper which presents the experiment in enormous detail but also in such a way that it can be quite confusing work one's way through it, especially in interpreting the results which are presented in a rather confusing and unclear way, However, the main problem I have with the paper is that it is not what it claims to be - a study of rural landscapes and their role in stress reduction - it is a study of PHOTOS of rural landscapes - a very different thing altogether. Nowhere is mention made in the abstract of this fact and there is no discussion about the reliability of photographs as surrogates for real landscapes - as has been done in many preference studies. As a consequence the study has no real point unless the idea is to use photographs and show them to stressed students as a way to cope with stressful conditions. If the idea is to suggest that people visit the countryside to recover from stress then the study would be hard to transfer into that recommendation. A better design might have been to show films of hectic urban environments in a VR kind of setting, followed by a film of a natural or rural scene - similar to what has been used to test forest bathing. Therefore I have to conclude that the paper has major weaknesses and needs to be reconsidered.

Reviewer 2 Report

tbc

General remarks:

Very interesting, valuable and well designed paper joining social, humanistic and formal levels of cultural landscape research. Abstract is clear and reflects the content of the paper, please write only about connection between rural landscapes and experiment with human subjects (I mean lines 14-17).

Introduction is also properly designed, with all needed references. Maybe it could be a bit shorten.

Section of Material and Methods is properly and comprehensively presented. The table 2 needs wider explanation – can you add any reference or criteria for dividing naturalness into three-level scale naturalness (strong - relatively strong – weak)? Description of the experiment also needs explanation. How many photos did You show to student? Each of presented (line 137-141)? It is quite confusing. There is any relation between the method with Scenic Beauty Estimation (Daniel, Boster 1976)? What?

The scope of the work is properly and comprehensively presented in the part of results. Discussion is really interesting and dividing it into section makes argument easy to  understood.

Conclusion also seems to be well done, answering points signalised in the Introduction. The language is sometimes quite rough, maybe it could be slightly improved.

Congratulations!

Detailed remarks:

Line 110 Table 1 – what do you mean by the heading “rural name”, I should be village name I think…

Figures with box-plots can be smaller.

Reviewer 3 Report

This is a topic I am interested in and I am pleased to read new work in this area. Unfortunately, I cannot recommend this paper is accepted, for two main reasons:

- The end of the results section needs serious reconsideration. At lines 334-345 you have four points (starting at number 4??), which attempt to briefly answer your final hypothesis. The text at the moment is confusing and seems unfinished. You need to complete this analysis fully, and show the results properly as you have with heart rate, blood pressure etc. It is not acceptable to show results in the way you have here. 

- The manuscript has not been read through properly. There are many spelling and grammar mistakes. This makes it confusing to read. Some examples from the introduction/methods/results:

    - Lines 22-26 "had significant different" does not make sense (this is repeated throughout the manuscript)

    - Line 71 Incorrect way of referencing in text

    - Line 78 poor grammar

    - Line 148 form -> from

    - Line 158 You are clearly missing words here.

    - Line 216 The graph shows blood pressure and heart rate were measured at T2, the figure says otherwise

    - Line 221 why is Concise Emotional State Scale in capitals? This is not the name of the scale

    - TMD formula has a comma at the end?

    - Line 254 do you mean preference, not performance?

    - Line 260 You repeatedly say "reached the 0.05 significance level". Please give the actual p-value.

    - Line 266 I personally find 'sCort' confusing - can you just say salivary cortisol instead? Seems a little unnecessary to shorten it

    - Line 273 p value is repeatedly shown as 'p=0.xxx>0.05'. You do not need to put >0.05 after it.

    - Line 323-325 There are numerous spelling/grammar mistakes in these sentences.

    - No y-axis on Figure 9.

    - Figure 9 caption does not make any sense and has a spelling error.

    - Line 238 belter? spelling mistake.

There are many other small mistakes throughout the manuscript. Please ensure it is read thoroughly. I believe the manuscript will be improved greatly if the full analysis is completed and described, and the writing mistakes are addressed.